# CSQ: Centered Symmetric Quantization for Extremely Low Bit Neural Networks

## Abstract

Recent advances in quantized neural networks (QNNs) are closing the performance gap with the full precision neural networks. However at very low precision (i.e., $\leq$ 3-bits), QNNs often still suffer significant performance degradation. The conventional uniform symmetric quantization scheme allocates unequal numbers of positive and negative quantization levels. We show that this asymmetry in the number of positive and negative quantization levels can result in significant quantization error and performance degradation at low precision. We propose and analyze a quantizer called *centered symmetric quantizer (CSQ)*, which preserves the symmetry of latent distribution by providing equal representations to the negative and positive sides of the distribution. We also propose a novel method to efficiently map CSQ to binarized neural network hardware using bitwise operations. Our analyses and experimental results using state-of-the-art quantization methods on ImageNet and CIFAR-10 show the importance of using CSQ for weight in place of the conventional quantization scheme at extremely low-bit precision (2~3 bits).

## 1 Introduction

Quantized neural networks (QNNs) (Krishnamoorthi, 2018; Esser et al., 2019; Lee et al., 2021; Li et al., 2021; Nagel et al., 2020) can reduce both computational complexity and memory requirement quite effectively over the full-precision (i.e., floating-point) version, and hence are commonly used for deployment. Recent state-of-the-art QNNs (Esser et al., 2019; Lee et al., 2021; Li et al., 2019) can achieve near-full-precision accuracy even at 3-bit for ImageNet classification. However, most of the existing quantization method target 3-bit or higher. At 2-bit there is not much room for optimization, even with non-uniform step-size quantization (Li et al., 2019).

While some previous work, *e.g.* (Li et al., 2019), treats 2-bit as ternary ($\{-1, 0, 1\}$), it wastes one quantization level. But to make the most of the 2-bit precision, the conventional 2-bit quantizer allocates two vs. one quantization levels to the negative side, causing a severe imbalance. As an alternative to the conventional quantizer, we propose *(zero-)centered symmetric quantizer (CSQ)*, which is most effective at low precision (2~3-bit) though the method itself is applicable to any precision. Through a set of analyses and experiments we show that CSQ improves QNN performance over the conventional linear quantizer (*CLQ*) in quantization-aware training (QAT).[1]

We note that the particular quantizer function we propose has been used in some previous work (Choi et al., 2018a; Boo et al., 2021; Gong et al., 2019; Lee et al., 2021; Chen et al., 2021). However, no previous work has proposed it explicitly nor provided any analysis on the effectiveness of such a scheme (see Section 2.3). Moreover, while low-precision CLQ can be efficiently implemented in hardware via multiple invocations of XNOR-popcount such as on BNN (Binarized Neural Network) hardware, doing the same for CSQ poses new challenges that have never been addressed.

In this paper we make the following contributions.

- We propose CSQ that uses perfectly symmetrical quantization levels with uniform step size. We provide analytical and empirical evidence showing that using CSQ for weight instead of CLQ improves performance of low-bit QNNs (2~3-bit).

---

[1] We use CSQ (and CLQ) to refer to both a quantizer and a quantization method.

- We propose a binary coding scheme and a mapping method for CSQ, which allows for an efficient hardware implementation of QNNs using bitwise operations on BNN hardware.

## 2 BACKGROUND AND RELATED WORK

### 2.1 QUANTIZATION PRIMER

A quantizer is a function from a real number $x$ to a discrete value or an integer $x_Q$. To train a network while simulating the effect of quantizer, one often maps the discrete value $x_Q$ back to a real value $\tilde{x}$. Instead of quantizing the entire range of inputs, clipping off the extreme values and only mapping the mid-range values uniformly is often more efficient. A generalized version of a quantizer used in Neural Networks can be defined as:

$$\tilde{x} = \text{clip}\left(\left\lfloor \frac{x}{s} \right\rceil, L, U\right) \cdot s \tag{1}$$

where $L$ and $U$ (the lower and upper bound) are the minimum and maximum integer values that $x_Q$ can take, and $\text{clip}(x, a, b) = \min(\max(x, a), b)$. Quantization using the above quantizer in (1) is called *uniform step-size quantization*, also called *linear quantization*. We do not consider non-uniform step-size quantization in this paper.

Based on the input range (i.e., whether input is signed vs. un-signed), quantizers are classified as symmetric vs. asymmetric quantizers. For the linear quantizer in (1), symmetric and asymmetric quantizers can be defined as follows, where $b$ is the number of bits used for representation.

Symmetric quantizer (for signed input): $\qquad L = -2^{b-1}, \qquad U = 2^{b-1} - 1 \qquad$ (2)

Asymmetric quantizer (for un-signed input): $\qquad L = 0, \qquad U = 2^b - 1 \qquad$ (3)

From here on we will refer to quantization method defined by (1) and (2) as the Conventional Linear Quantizer (CLQ).

### 2.2 PREVIOUS WORK ON DNN QUANTIZATION

Most of the previous works on DNN quantization (Jung et al., 2019; Choi et al., 2018b; Esser et al., 2019) use essentially the same linear quantizer, but each work proposes a slightly different way to optimize quantizer functions. Often the differences come from different formulations of the quantizer function. For instance, QIL (Jung et al., 2019) optimizes quantization boundary, PACT (Choi et al., 2018b) optimizes a clipping parameter to train the quantized activations, and LSQ(Esser et al., 2019) optimizes the step size. We are not aware of any previous work explicitly proposing a new set of quantization levels for uniform quantization. Though the formulation of a quantizer function varies, in the end they all assume the same integer hardware that expects 2's complement number representation.

### 2.3 SIMILAR APPROACHES

The idea of using a zero centered quantizer is also seen in (Choi et al., 2018a; Boo et al., 2021; Gong et al., 2019; Lee et al., 2021; Chen et al., 2021). All of these works present a novel training method for their quantizer. However, none of these works analyze zero-centered quantization or present it as a source of performance gain. Furthermore, to the best of our knowledge, no previous work has attempted to address the realization of zero-centered quantization on hardware. Realizing zero-centered quantization on hardware can be a challenging task, because it introduces new quantization points which can not be represented using 2's complement number representation.

## 3 CENTERED SYMMETRIC QUANTIZATION (CSQ)

### 3.1 EXACT ZERO REPRESENTATION VS. PERFECT SYMMETRY

Any linear quantization to $b$-bit precision results in $2^b$ quantization levels. Therefore, inclusion of zero among the quantization points, naturally results in an asymmetry in the number of positive and

Table 1: Quantization levels by representation (2-bit example)

| Case | Quantization levels | Representation |
|------|--------------------|-----------------|
| 1 | $\{-2, -1, 0, 1\}$ | CLQ |
| 2 | $\{-1, 0, 1, 2\}$ | CLQ-alternative |
| 3 | $\{-1, 0, 1\}$ | RSQ (Ternary) |
| 4 | $\{-2, -1, 0, 1, 2\}$ | ESQ* |
| 5 | $\{-2, -1, 1, 2\}$ | NSQ |
| 6 | $\{-1.5, -0.5, 0.5, 1.5\}$ | CSQ |
| 7 | $\{-3, -1, 1, 3\}$ | CSQ (scaled by 2) |

*Note: ESQ requires more than 2 bits.

negative quantization levels. While exact representation of zero may be important due to common operations like zero padding (Krishnamoorthi, 2018), it is also true that the asymmetry among positive and negative quantization levels grows large as precision becomes low. Thus one of our aims in this paper is to explore the trade-off between exact zero representation and perfect symmetry in the context of weight quantization of neural networks.

### 3.2 ALTERNATIVES

From the perspective of quantization levels, one can consider the following quantization schemes (see Table 1). **Conventional Linear Quantization (CLQ)** uses the quantization levels represented by 2's complement, which is also used by most of the previous uniform step-size quantizers, *e.g.* (1). Alternatively, the asymmetry between the positive and negative sides can be reversed; i.e., by setting $L = -2^{b-1} + 1$ and $U = 2^{b-1}$. **Reduced Symmetric Quantization (RSQ)** uses one less quantization level, thereby achieving both exact zero representation and perfect symmetry; i.e., $L = -2^{b-1} + 1$ and $U = 2^{b-1} - 1$. This scheme wastes one quantization level and is expected to result in inferior performance. **Extended Symmetric Quantization (ESQ)**, on the other hand, uses one more quantization level to achieve zero representation and symmetry; i.e., $L = -2^{b-1}$ and $U = 2^{b-1}$. However, since ESQ requires more than $b$-bit precision, it is not feasible for practical deployment, but included here for comparison. **Non-uniform Symmetric Quantization (NSQ)** achieves both exact zero representation and perfect symmetry with exactly $2^b$ quantization levels, but the step size is not uniform, leading to a completely different quantizer function and QAT methods. Since NSQ is non-uniform quantization, we do not consider it for the scope of this paper. **Centered Symmetric Quantization (CSQ)** stipulates *uniform step size* and *perfect symmetry* between the positive and negative sides, while compromising on the exact representation of zero. It can also be represented using integers only by scaling all quantization levels by 2 as shown in the last row of the table. For the remainder of the paper we focus on CSQ and CLQ, as they are the most practical.

### 3.3 QUANTIZER FUNCTION FOR CSQ

We define the quantizer for CSQ as follows:

$$\dot{v} = \left\lfloor \frac{v}{s} + 0.5 \right\rceil - 0.5 \tag{4}$$

$$\bar{v} = \text{clip}\left(\dot{v}, -Q, Q\right) \tag{5}$$

$$\hat{v} = \bar{v} \times s \tag{6}$$

where $v$ is any input value, $s$ is the step size, and $Q = 2^{b-1} - 0.5$ with $b$ being the quantization precision (i.e., the number of bits). As usual, $\lfloor \cdot \rceil$ is the round operation, and $\text{clip}(x, a, b) = \min(\max(x, a), b)$. Here $\bar{v}$ is not an integer. Nevertheless, it is an exact value that can be represented by the $b$-bit CSQ format. Moreover, our proposed CSQ format permits efficient hardware and software realizations (see Section 5). Therefore $\bar{v}$ represents the value that is computed by $b$-bit hardware. Finally, $\hat{v}$ is the scaled-back version of $\bar{v}$, defined and used for the purpose of training. The proposed quantizer provides equal representation for the positive and negative sides of the input distribution. Figure 1a shows the 2-bit quantizer functions for conventional linear quantization and our CSQ. Figure 1b shares the gradient for step size which is a quantization parameter. Detailed training method is described in Appendix A.

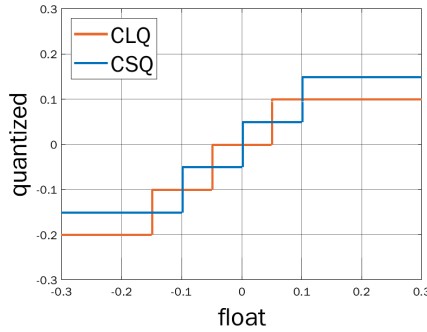
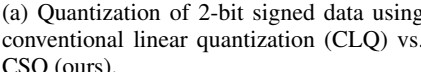

(a) Quantization of 2-bit signed data using conventional linear quantization (CLQ) vs. CSQ (ours).

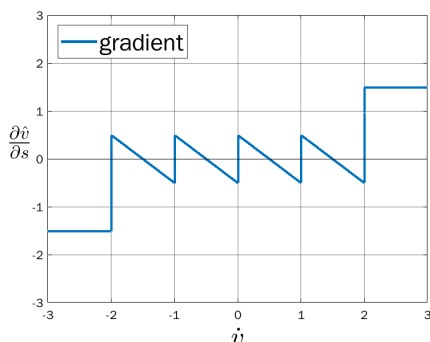

(b) The gradient of the CSQ quantization output w.r.t. step size.

Figure 1: CSQ quantizer and gradient.

Table 2: List of values in the product term (2-bit example, unsigned activation)

|  | W-A: $\text{CLQ}_s$-$\text{CLQ}_u$ | W-A: CSQ-$\text{CLQ}_u$ |
|---|---|---|
| W (signed) | $\{-2, -1, 0, 1\}$ | $\{-1.5, -0.5, 0.5, 1.5\}$ |
| A (unsigned) | $\{0, 1, 2, 3\}$ | $\{0, 1, 2, 3\}$ |
| $W \cdot A$ | $\{-6, -4, -3, -2, 1, 0, 1, 2, 3\}$ | $\{-4.5, -3, -1.5, -1, -0.5, 0, 0.5, 1, 1.5, 3, 4.5\}$ |

## 4 ANALYSIS OF CSQ

### 4.1 IMPROVED REPRESENTATIONAL CAPACITY OF CSQ

Even though both CSQ and CLQ have the same number of quantization levels, and therefore the same representational capacity on the operand level, multiplication result, or the product of weight and activation, may have different representational capacity depending on the choice of quantizer. We compare the representational capacity of the product of weight and activation, when using CSQ vs. CLQ for weight. For activation quantization we consider unsigned CLQ (denoted by $\text{CLQ}_u$) for unsigned activation, and either CSQ or signed CLQ (denoted by $\text{CLQ}_s$) for signed activation. We start by defining the range of quantization levels of each scheme:

$$\text{CLQ}_u \sim \{0, 1, \cdots, 2^b - 1\} \tag{7}$$

$$\text{CLQ}_s \sim \{-2^{b-1}, -2^{b-1} + 1, \cdots, 2^{b-1} - 1\} \tag{8}$$

$$\text{CSQ} \sim \{-2^{b-1} + 0.5, -2^{b-1} + 1.5, \cdots, 2^{b-1} - 0.5\} \tag{9}$$

where $b$ is the precision. It should be noted that CSQ can only be signed. Table 2 shows the range of weights, activations and their product for 2-bit precision using different quantization methods.

Table 3 shows that using CSQ for weight quantization almost always increases the representational capacity of quantized multiplication. The only exception is 2-bit precision for signed activation where it remains the same. Given that improving representational capacity is crucial for increasing the overall performance of QNNs (Liu et al., 2018), our analysis suggests that CSQ has a definite advantage over CLQ for quantizing weights. Furthermore, the distribution of product resembles non-uniform quantization which provides higher accuracy. For more details refer to Appendix C.

In the case of signed activation, our result in Table 3 suggests that using $\text{CLQ}_s$ for activation and CSQ for weight provides the highest representational capacity.

### 4.2 CSQ AND CLQ AS DIFFERENT ZERO-POINTS OF AFFINE QUANTIZATION

In this section we analyze and compare CSQ and CLQ using a more general framework of affine quantization, which is defined as follows (Krishnamoorthi, 2018):

$$x_{int} = \left\lfloor \frac{x}{s} \right\rceil + z, \quad \bar{x} = \text{clip}(x_{int}, 0, 2^b - 1), \quad \hat{x} = s(\bar{x} - z) \tag{10}$$

Table 3: Number of unique values in the product term

| #bits | Unsigned Activation (W-A) | | Signed Activation (W-A) | | |
|---|---|---|---|---|---|
| | $\text{CLQ}_s$-$\text{CLQ}_u$ | $\text{CSQ}$-$\text{CLQ}_u$ | $\text{CLQ}_s$-$\text{CLQ}_s$ | $\text{CSQ}$-$\text{CLQ}_s$ | $\text{CSQ}$-$\text{CSQ}$ |
| 2-bit | 9 | 11 (+22.2%) | 6 | 9 (+50.0%) | 6 (0.0%) |
| 3-bit | 35 | 43 (+22.9%) | 18 | 31 (+72.2%) | 20 (+11.0%) |
| 4-bit | 120 | 155 (+29.2%) | 60 | 105 (+75.0%) | 66 (+10.0%) |

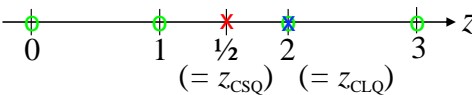

Figure 2: Zero-point for CLQ and CSQ at 2-bit precision. Affine quantizer allows integer zero-point only (shown in green circles). In relaxed affine quantizer, zero-point can take any real value.

where $z$ is called *zero-point*, and must be an integer since the other terms on both sides are all integers. However if we relax the zero-point to be any real value, CLQ and CSQ can both be considered as a special case of (relaxed) affine quantizer. Then the zero-point for CLQ ($z_{\text{CLQ}}$) and CSQ ($z_{\text{CSQ}}$) can be defined as follows:

$$z_{\text{CLQ}} = 2^{b-1}, \quad z_{\text{CSQ}} = 2^{b-1} - 0.5 \tag{11}$$

For instance, at $b = 2$, the term $\left\lfloor \frac{x}{s} \right\rceil$ has the range of $\{-2, -1, 0, 1\}$ in (signed) CLQ. Thus to satisfy (10), we have $z_{\text{CLQ}} = 2$.

We can consider four cases depending on the restriction on zero-point (ZP): (i) ZP can take any real value (i.e., relaxed affine quantizer), (ii) ZP can take any integer value (i.e., affine quantizer), (iii) ZP is fixed to $z_{\text{CLQ}}$, and (iv) ZP is fixed to $z_{\text{CSQ}}$. Among these, CLQ is a special case of integer zero-point (affine quantizer) while CSQ is not (see Figure 2). At the same time, there is a hardware overhead for (i) and (ii), though integer zero-point can be implemented more efficiently than real-value zero-point. We show in Section 5 that CSQ can be implemented as efficiently as CLQ.

The above view provides us with a methodology to somewhat objectively compare CSQ and CLQ by the distance of the optimal real-value ZP ($z^*$) to $z_{\text{CSQ}}$ vs. $z_{\text{CLQ}}$; of which we present our results in Section 6.5. Also, it follows from (11) that the difference between $z_{\text{CLQ}}$ and $z_{\text{CSQ}}$, relative to the entire quantization range, diminishes exponentially as $b$ increases (see Appendix H), suggesting that the performance difference between CLQ and CSQ will be negligible for large $b$.

## 5 EFFICIENT REALIZATION OF CSQ

The high efficiency of BNN hardware comes from the bit-parallel processing (i.e., bitwise XNOR) of large arrays, which can be extended to efficient processing of very low-precision QNNs by interchanging the order of summation between precision and vector dimensions as observed by Zhou et al. (2016).

The challenge is how to extend the bitwise-operation based inner-product computation method to CSQ. Since there are multiple combinations, here we show the result for the case of a $\text{CLQ}_u$ vector $\mathbf{x}$ and a CSQ vector $\mathbf{v}$, each of which is $N$-dimensional and $n$-bit.

An $n$-bit binary number $a_{n-1}a_{n-2}...a_0$ ($n \geq 2$) represents the following value: $A_b = \sum_{i=0}^{n-1} a_i 2^i$. We define the CSQ number format as exemplified in Table 4, which leads to the following equalities (the second equality is not obvious, but is correct).

$$A_{csq} = \sum_{i=0}^{n-1} a_i 2^i - (2^n - 1)/2 = \sum_{i=0}^{n-1} (-1)^{a_i+1} 2^{i-1} \tag{12}$$

To avoid dealing with fractional numbers, let us use the $2\times$ scaled version of CSQ (see Table 1). Then,

$$A_{csq2} = \sum_{i=0}^{n-1} (-1)^{a_i+1} 2^i, \tag{13}$$

Table 4: Comparison of number representations: CLQ vs. CSQ (2-bit example)

| 2-bit binary | $CLQ_s$ | $CLQ_u$ | CSQ |
|---|---|---|---|
| 00 | 0 | 0 | $-1.5$ |
| 01 | 1 | 1 | $-0.5$ |
| 10 | $-2$ | 2 | $0.5$ |
| 11 | $-1$ | 3 | $1.5$ |

which has a very similar mathematical structure as a CLQ number, allowing us to use the same trick of changing the order of precision ($n$) and vector dimension ($N$) as in the inner-product computation of two CLQ numbers. Finally, we arrive at the following inner-product computation method (for more detail, see Appendix I):

$$\mathbf{v} \cdot \mathbf{x} = (\mathbf{v}_H \cdot \mathbf{x}_H << 2) + (\mathbf{v}_H \cdot \mathbf{x}_L << 1) + (\mathbf{v}_L \cdot \mathbf{x}_H << 1) + \mathbf{v}_L \cdot \mathbf{x}_L \tag{14}$$

which is shown for the 2-bit case ($n = 2$). $\mathbf{x}_H$ and $\mathbf{x}_L$ (similarly for $\mathbf{v}_H$ and $\mathbf{v}_L$) are the $N$-dimensional bit-vectors of $\mathbf{x}$ containing only the higher and lower bits only, respectively, and $<<$ is the bitwise shift-left operation. Each product on the right-hand side of (14) can be computed on BNN hardware in a single cycle. Thus $\mathbf{v} \cdot \mathbf{x}$ can be computed in four cycles, using an additional adder/accumulator. It is worth mentioning that the same method as illustrated in (14) is also used when computing the inner-product of two CLQ vectors.

Now for the inner-product of two bit-vectors (*e.g.* $\mathbf{v}_H \cdot \mathbf{x}_H$) we can use the same structure of a bitwise operation followed by popcount, with a slight variation.

$$\mathbf{v}_{csq2} \cdot \mathbf{x}_{clq_u} = 2 \cdot \text{popcount}(\text{AND}(\mathbf{v}_{csq2}, \mathbf{x}_{clq_u})) - \text{popcount}(\mathbf{x}_{clq_u}) \tag{15}$$

We subtract $\text{popcount}(\mathbf{x}_{clq_u})$, since the bits of the CSQ bit-vector corresponding to the zero elements of the CLQ bit-vector must be ignored. Note that BNN hardware, *e.g.* (Umuroglu et al., 2017), relies on the same method for the inner-product computation of two bit-vectors, except that we use AND instead of XNOR and to subtract a popcount value, we need an additional popcount operation. However, accelerating QNNs with the CLQ format also requires AND operations in the exact same manner, and the additional popcount operation does not increase the hardware cost. Thus the bit-vector-level complexity of using CSQ is nearly the same as that of using CLQ. Since there is no distinction at the vector level, the inner-product operation with CSQ can be implemented as efficiently as with CLQ.

## 6 EXPERIMENTS

### 6.1 EXPERIMENTAL SETUP

We conduct experiments on CIFAR-10 (Krizhevsky et al., 2009) and ImageNet (Russakovsky et al., 2015) using ResNet-18, ResNet-20, ResNet-34 (He et al., 2016) and MobileNet-v2 (Sandler et al., 2018). All quantized models on ImageNet are initialized with the weights of pretrained full-precision model of the same network. The first and last layers are kept at 8-bit precision. Other than convolution and fully connected layers, all the other layers, *e.g.* batch norm, are kept in full precision.

For ImageNet experiments we use the training recipe of Esser et al. (2019). We use stochastic gradient descent (SGD) optimizer, with 0.9 momentum, cosine learning rate decay (Loshchilov & Hutter, 2016) without restarts, and the initial learning rate of 0.01. Weight decay is $0.25 \times 10^{-4}$ for 2-bit, $0.5 \times 10^{-4}$ for 3-bit, and $10^{-4}$ for 4-bit quantization. The quantized models are fine-tuned for 90 epochs. For ResNet-18 ImageNet experiments, we train the full-precision model ourselves from scratch, using the same experimental setup except 0.1 initial learning rate and $10^{-4}$ weight decay.

For CIFAR-10 experiments the weights for quantized models are trained from scratch. We use the same setting as with ImageNet except the following: the initial learning rate is 0.1, weight decay is $10^{-4}$, and each model is trained for 300 epochs. All experiments on CIFAR-10 are conducted using ResNet-20 network. All ImageNet and CIFAR-10 models were implemented in PyTorch.

Table 5: Comparison of CLQ and CSQ (ours) on ResNet-20 for CIFAR-10.

| | Top-1 Accuracy @ Precision | | |
|---|---|---|---|
| **Network** | ResNet-20 | | |
| | *Full Precision: 91.22* | | |
| **Precision (W/A)** | 2/2 | 3/3 | 4/4 |
| CLQ (LSQ) | 89.09±0.11 | 90.70±0.18 | **91.07±0.13** |
| CSQ (ours) | **89.46±0.225** | **90.80±0.12** | 90.96±0.25 |

Table 6: Comparison of CLQ, CSQ and other quantization-aware training methods on ImageNet. The CLQ case also represents LSQ (Esser et al., 2019). MobileNet-v2 2-bit case did not converge.

| | Top-1 Accuracy @ Precision | | | | | | | | |
|---|---|---|---|---|---|---|---|---|---|
| **Network** | ResNet-18 | | | ResNet-34 | | | MobileNet-v2 | | |
| | *Full Precision: 70.58* | | | *Full Precision: 73.31* | | | *Full Precision: 71.88* | | |
| **Precision (W/A)** | 2/2 | 3/3 | 4/4 | 2/2 | 3/3 | 4/4 | 2/2 | 3/3 | 4/4 |
| PACT | 64.40 | 68.10 | 69.20 | - | - | - | - | - | - |
| LQ-Nets | 64.90 | 68.20 | 69.30 | 69.80 | 71.90 | - | - | - | - |
| QIL | 65.70 | 69.20 | 70.10 | 70.60 | 73.10 | 73.70 | - | - | - |
| CLQ (LSQ) | 66.59 | 69.38 | 70.52 | 70.56 | 73.21 | 73.82 | - | 60.41 | 66.82 |
| LSQ+ | 66.80 | 69.40 | **70.80** | - | - | - | - | - | - |
| CSQ (ours) | **66.92** | **69.48** | 70.63 | **70.82** | **73.29** | **74.01** | - | **60.89** | **66.98** |

## 6.2 CIFAR-10 RESULT

We compare our method with CLQ using a state-of-the-art QAT method by Esser et al. (2019). Each case is repeated five times; mean ± std. dev. is reported for each case. The results are summarized in Table 5, which confirms that our proposed method performs significantly better than CLQ at 2-bit, while at 3- and 4-bit, there is little difference in performance between CLQ and CSQ.

## 6.3 IMAGENET RESULT

We compare CLQ vs. CSQ on ImageNet classification using the standard models such as ResNet and MobileNet. We also compare our method with PACT (Choi et al., 2018b), LQ-Nets (Zhang et al., 2018), QIL (Jung et al., 2019) and LSQ+ (Bhalgat et al., 2020). Since we use the training method by LSQ (Esser et al., 2019), the CLQ case also represents the previous work (LSQ). However, LSQ use pre-activation ResNet (He et al., 2016) which has higher performance than the standard ResNet architecture, and their trained models are not available. Therefore for a fair comparison, we have implemented LSQ ourselves, and use it as the baseline. For CSQ we use the same model and replace only the weight quantizer with CSQ, except in the first and last layers which are quantized to 8-bit CLQ. LSQ+ (Bhalgat et al., 2020) uses affine quantizer for activation that allows floating-point zero-point. However, ResNet and MobileNet use ReLU activation which results in unsigned activation which can not fully utilize the capability of affine quantizer. Therefore, we do not use affine quantizer for activations in our experiments. For comparison with LSQ+ we use their reported results. Lee et al. (2021) is another recent work that presents a training method for QNNs, that outperforms LSQ. However, they already use a quantizer than results in zero-centered quantization levels, similar to the one proposed in this paper. Therefore, we do not compare our results with them.

The results are summarized in Table 6 which shows that CSQ outperforms CLQ for all cases. We can observe that the performance gain by our method diminishes as precision increases. At 2- and 3-bit precision CSQ outperforms LSQ+ as well. We attribute this gain in performance to the improved representational capacity of CSQ, discussed in Section 4.1, and reduced quantization error shown in Section 6.4.

## 6.4 QUANTIZATION ERROR USING LSQ

We study the quantization error using CLQ and CSQ. We use full precision weights trained on ImageNet, which are also used to initialize weights for quantization-aware training. We compare

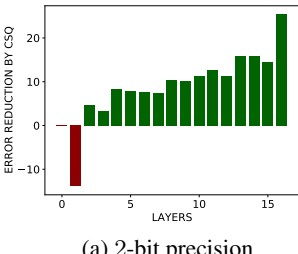 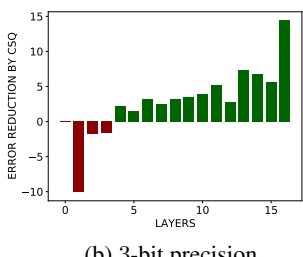 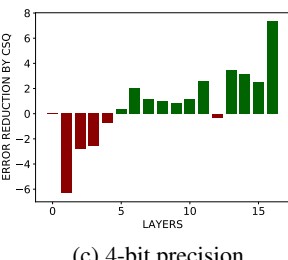

(a) 2-bit precision       (b) 3-bit precision       (c) 4-bit precision

Figure 3: Percentage error reduction using CSQ. The results are shown for ResNet-18 trained on ImageNet.

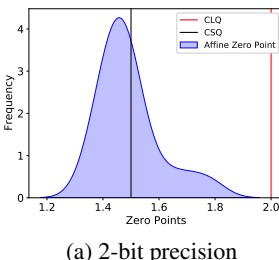 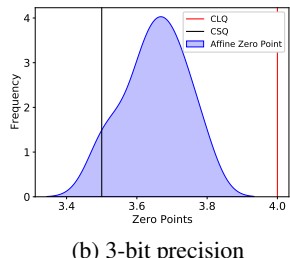 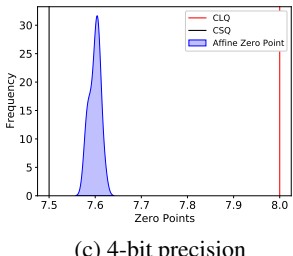

(a) 2-bit precision       (b) 3-bit precision       (c) 4-bit precision

Figure 4: Distribution of optimal zero-point values of relaxed affine quantizer, across layers. The graphs also show the zero-points for CSQ and CLQ, represented as vertical lines.

mean square error $\langle (w - w_q)^2 \rangle$ between CLQ and CSQ, where $w$ is the full precision weight and $w_q$ is the quantized weight. For each experiment we find a step size ($s$) that minimizes each error metric, by an exhaustive search. We report percentage error reduction by CSQ for each layer. which we define as: $\text{error\_reduction}(\%) = \frac{(E_{CLQ} - E_{CSQ})}{E_{CLQ}} \times 100$, where $E_{CLQ}$ and $E_{CSQ}$ are the quantization errors using CLQ and CSQ, respectively. The results are presented in Figure 3.

It can be seen that CSQ reduces the overall quantization error compared to CLQ. At 2-bit precision CSQ provides a clear advantage over CLQ. At 3-bit precision CSQ is still significantly better than CLQ but the improvement in quantization error using CSQ is less comapred to 2-bit. Finally at 4-bit precision the improvement using CSQ seems much smaller than 2 and 3-bit precision. Additionally CLQ provides significantly results in some layers compared to CSQ. Based on these results we can conclude that CSQ provides significant advanatge at 2-bit precision, which diminishes as we increase the precision.

## 6.5 ANALYZING CSQ USING AFFINE QUANTIZER

To analyze the effectiveness of CSQ, we train ResNet-20 for 2, 3 and 4-bit precision on CIFAR-10, using affine quantization defined in (10). The latent weights for quantized model, are initialized using pre-trained full precision weights. We initialize and train the step size using LSQ (Esser et al., 2019), similar to experiments in Section 6.2 and Section 6.3. Zero-point is relaxed to be a real value and initialized to $z_{\text{CLQ}}$ from (11). We choose the initialization biased towards CLQ to show that zero-point converges closer to CSQ irrespective of initialization.

The results are shown in Figure 4. It can be seen that despite our initialization being biased towards CLQ, the zero-point distribution is always closer to CSQ than CLQ. Considering that the zero point distribution moves away from CSQ as the precision increases, the results suggest that there is an inverse relation between the effectiveness of CSQ and precision of the network, which is also consistent with our analytical result in Section 4.2, that CSQ is most effective at low precision (i.e., 2-bit) and loses effectiveness as we increase precision. The result in Figure 4 also indicates that CSQ provides significantly better approximation of affine quantization than CLQ.

Table 7: Matrix multiplication runtime (size: $16384 \times 16384$) on Nvidia RTX 2080 TI

| Kernel | Runtime (ms) | Relative Speed |
|---|---|---|
| cuBLAS | $579.81 \pm 9.16$ | 1 |
| 1-bit | $85.44 \pm 11.99$ | 6.8 |
| 2-bit CSQ | $197.85 \pm 6.82$ | 2.9 |
| 2-bit CLQ | $197.85 \pm 10.20$ | 2.9 |

## 6.6 GPU IMPLEMENTATION RESULTS

To see the speedup on GPU, we have implemented three custom kernels that use concatenation and bitwise operations to compute matrix multiplication (see Appendix I.2). We compare these custom kernels with cuBLAS. The first kernel is 1-bit CSQ, which is the same as 1-bit BNN. The second and third kernels are 2-bit, each using CSQ and CLQ.

The result is summarized in Table 7, which is the average of five runs. We observe exactly the same speed between CSQ and CLQ, which demonstrates that our CSQ is as efficient as the conventional binary representation on GPU. Note that apart from the main computation kernel, our CSQ is exactly the same as CLQ (such as memory requirement) and therefore has the same overall runtime as CLQ. Also our 2-bit CLQ is 2.9 times faster than cuBLAS while being only 2.3 times slower than 1-bit.

## 6.7 DISCUSSION

We can see from experimental results and analyses that CSQ provides significant advantage at 2-bit precision. It is also evident that CSQ loses significance as we increase precision. The question is, **"what is the exact precision where CSQ loses its superiority to CLQ ?"**. Experimental results and quantization error experiments in Section 6.4 demonstrate that superiority of CSQ is significantly reduced at 4-bit precision. However representational capacity analysis in Section 4.1 and affine quantization experiments in Section 6.5 suggest that CSQ provides significant advantage at 4-bit precision as well.

We know that 4-bit precision does not suffer from performance degradation using Quantization-Aware Training (QAT). In fact with current state-of-the-art 4-bit precision achieves similar or even higher accuracy than full-precision networks for ResNet. This can also be seen in our experimental results in Table 6. In other words, for ResNet-18 4-bit precision can provide more than sufficient representation irrespective of quantization method and lack of representation capacity does not remain a bottleneck in performance. That is why at 4-bit precision we observe very similar results whether we use CSQ or CLQ. Based on this we conclude that CSQ achieves its limit at 4-bit precision for ResNet and beyond 4-bit precision CLQ and CSQ should perform similarly. While the exact precision threshold cannot be generalized to other networks or datasets (*e.g.* CSQ outperforms CLQ on ResNet-18 and ResNet-34 on ImageNet but does not provide the same superiority on ResNet-20 on CIFAR-10), the criteria to find the threshold are generalizable; that is, CSQ is superior while representation capacity is the bottleneck.

Finally, though we have used LSQ only (Esser et al., 2019) as the quantization method in our experiments, our quantizer can be applied with any quantization method. In Appendix D we present our PTQ (post-training quantization) result using another state-of-the-art quantization method (Li et al., 2021).

## 7 CONCLUSION

In this paper we provided an in-depth analysis of CSQ for extreme low-bit quantization, which is both completely symmetric around zero and trainable using existing linear quantization methods. Our analyses and experimental results using state-of-the-art quantization methods with CIFAR-10 and ImageNet datasets show that a simple change of quantization levels can result in significant performance improvement for extremely low bit quantized neural networks ($\leq 4$ bits). We also show that CSQ can be realized efficiently on BNN hardware and GPUs. Considering there are very few previous works targeting 2-bit network performance, CSQ can be a very useful tool for optimizing extreme low-precision neural networks for deployment.

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

# A    OUR TRAINING METHOD

We have employed the method presented in Esser et al. (2019) to train the quantization parameters. To optimize the step size $s$ using gradient descent, we use the following derivative formula.

$$\frac{\partial \hat{v}}{\partial s} = \begin{cases} -v/s + \dot{v} & \text{if } -Q < \dot{v} < Q \\ -Q & \text{if } \dot{v} \leq -Q \\ Q & \text{if } \dot{v} \geq Q \end{cases} \tag{16}$$

Then computing the loss gradient w.r.t. step size is straightforward. Similar to gradient scaling in (Esser et al., 2019), the gradient of step size is scaled by factor $g = 1/\sqrt{N_W 2^p}$, where $N_W$ is the number of weight parameters. The weights are initialized as $2\langle |v| \rangle / \sqrt{Q}$, where $\langle . \rangle$ represents the notation for mean of a distribution. Figure 1b shares the gradient of step size parameter at 2-bit precision. Our training method is based on LSQ (Esser et al., 2019) but it should be noted that CSQ can be used with any training method.

# B    LIMITAIONS OF AFFINE QUANTIZER

We have defined affine quantizer by Krishnamoorthi (2018) in (10). Another variant of affine quantizer has also been proposed by Bhalgat et al. (2020):

$$\bar{x} = \left\lfloor \text{clip}\left(\frac{x - z}{s}, n, p\right) \right\rceil, \quad \hat{x} = s \cdot \bar{x} + z \tag{17}$$

where $n$ and $p$ are clipping intervals. Bhalgat et al. (2020) does not restrict their zero-point to integer values. This approach can lead to significant hardware overhead. To tackle this problem they propose the affine quantizer for activations only, in which case zero-point can be implemented simply as a bias term, since

$$\hat{w} \cdot \hat{x} = (\bar{w} \cdot s_w)(\bar{x} \cdot s_x + z) = \bar{w}\bar{x}s_w s_x + \underbrace{z s_w \bar{w}}_{\text{bias}} \tag{18}$$

and $\bar{w}$ is known in advance. This resolves the hardware overhead problem of affine quantization for activation. However, the same method cannot be applied to weight because it would involve pre-computing $\bar{x}$ in advance, which is impossible.

Affine quantization for weights still results in some hardware overhead as the the zero-point of weights quantizer can not be implemented as the bias. This can be shown as:

$$\hat{w}\hat{x} = (\bar{w} \times s_w + z_w)(\bar{x} \times s_x + z_x) = \bar{w}\bar{x}s_w s_x + \underbrace{z_x s_w \bar{w} + z_w z_x}_{\text{bias}} + \underbrace{z_w s_x \bar{x}}_{\text{overhead}} \tag{19}$$

This is because $\bar{x}$ can not be pre-computed as it will change with each input. Similarly weight quantization overhead is shown by (Krishnamoorthi, 2018) as:

$$y(k, l, n) = s_w s_x \text{conv}\left(\bar{w}(k, l, m; n) - z_w, \bar{x}(k, l, m) - z_x\right) \tag{20}$$

$$y(k, l, n) = \text{conv}\left(\bar{w}(k, l, m; n), \bar{x}(k, l, m)\right) - z_w \sum_{k=0}^{K-1} \sum_{l=0}^{K-1} \sum_{m=0}^{N-1} \bar{x}(k, l, m) \tag{21}$$

$$- z_x \sum_{k=0}^{K-1} \sum_{l=0}^{K-1} \sum_{m=0}^{N-1} \bar{w}(k, l, m; n) + z_x z_w \tag{22}$$

Considering these limitations of affine quantizer we have proposed CSQ as an approximation of affine quantizer which can be efficiently realized on hardware.

# C    MORE ON REPRESENTATIONAL CAPACITY

In Section 4.1 we have shown that CSQ improves the representational capacity compared to CLQ. We have considered using signed, as well as unsigned activations. The most common activation

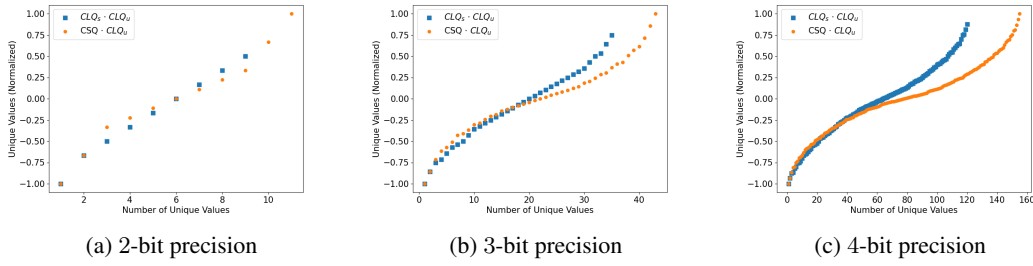

Figure 5: Distribution of quantized product values when using CSQ vs. $CLQ_s$ for weight, and $CLQ_u$ for unsigned activation.

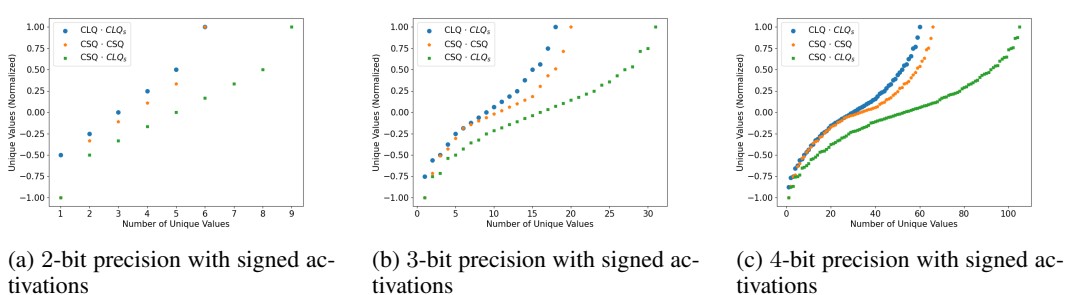

(a) 2-bit precision with signed activations

(b) 3-bit precision with signed activations

(c) 4-bit precision with signed activations

Figure 6: Representational capacity of quantized product with signed activations, using different quantization methods for 2, 3 and 4-bit precision.

function is ReLU, which produces unsigned activations. In case of signed activations such as PReLU and h-swish, affine quantization (Krishnamoorthi, 2018; Bhalgat et al., 2020) provides the best representation. However, affine quantizer can also be represented by unsigned CLQ before applying zero-points. Therefore unsigned CLQ for activations encapsulates affine quantizer as well. We also compare the representational capacity of quantized operation using symmetric activation range. Such an activation is unusual as the activation function introduces some non-linearity. However, it is possible in some cases e.g output of Tanh activation. In this case the activation is quantized using signed CLQ or CSQ. We show the normalized distribution of product with unsigned activations in Figure 5. It can be observed that the distribution of product values when using CSQ for weight is perfectly symmetric, and densely populated in the middle of the output range while being sparse near the boundary. This bell-shaped distribution better resembles layer activations, which is often exploited by non-uniform quantization schemes (Li et al., 2019; Lee et al., 2017) to achieve higher quantization performance. On the other hand, the product values when using CLQ for weight, are more uniformly distributed for the entire output range, with an exceptional outlier in the negative range and no matching positive outlier, introducing asymmetry to the output as well.

Figure 6 shows the normalized distribution of product with signed activation. It can be seen that using CSQ for weights and CLQ for activations provides significantly higher representational capacity compared to using CLQ for weights or CSQ for activations.

# D  POST TRAINING QUANTIZATION RESULTS (COMPARISON WITH BRECQ)

BRECQ (Li et al., 2021) uses affine quantizer with integer zero-point, which extends the linear quantizer in a different direction than our CSQ. Thus it is very interesting to see how our method compares with BRECQ. Note that integer zero-point requires some additional hardware overhead whereas CSQ does not. At the same time, the two schemes are orthogonal in the sense that one can combine both schemes, viz. integer zero-point and CSQ, so that zero-point take any integer or half-integer value, at the cost of some hardware overhead. Here we compare only CSQ vs. BRECQ,

Table 8: Comparison of CSQ with affine quantizer using BRECQ on ImageNet.

| | | Accuracy | | |
|---|---|---|---|---|
| **Network** | **Method** | 2/32 | 3/32 | 4/32 |
| ResNet-18 | | *Full Precision: 71.08* | | |
| | BRECQ | 66.60 | **69.82** | **70.64** |
| | BRECQ-CSQ (ours) | **66.93** | 69.81 | *70.62* |
| ResNet-50 | | *Full Precision: 77.00* | | |
| | BRECQ | 72.16 | **75.68** | **76.41** |
| | BRECQ-CSQ (ours) | **72.24** | 75.52 | 76.35 |

but not the combination. Since we have proposed CSQ for weight quantization, we only quantize the weights in BRECQ experiments.

To implement CSQ we simply fix the zero point value to CSQ which is shown as $Z_{CSQ}$ in (11). The shared results may be different from reported results. The results have been reproduced using the officially shared code. We share our results on ResNet-18 and ResNet-34. We used channel-wise quantization and all the hyper-parameters are same as shown in Li et al. (2021). Table 8 shows the experimental results using BRECQ on ImageNet data. It can be seen that at 2-bit precision CSQ outperforms affine quantizer. This is especially interesting because unlike affine quantization, CSQ does not incur any hardware overhead. At 3 and 4-bit precision, affine quantizer gives better performance. However, CSQ also provides competitive results. This shows that CSQ is a very strong approximation of (relaxed) affine quantizer.

## E    ADDITIONAL EXPERIMENTS USING KNOWLEDGE DISTILLATION

To further demonstrate the generalization-ability of CSQ, we conduct some experiments with knowledge distillation (Hinton et al., 2015) loss. We use LSQ (Esser et al., 2019) for training the quantization parameters and show that CSQ performs superior to CLQ at low precision even when we use advanced training methods such as knowledge distillation over state-of-the-art quantization-aware training method.

The experimental methodology and training setups are the same as described in Section 6.1, except that we use a weight decay of 0.25e-4 and knowledge distillation loss for all experiments. For knowledge distillation loss we set temperature as 1 and give equal weight to the standard loss and the distillation loss following (Esser et al., 2019). Our experimental results in Table 9 show that at extremely low precision, i.e., at 2- and 3-bit, CSQ outperforms CLQ. This is consistent with experimental results in Table 6. We have already discussed in Section 6.7 why CSQ may lose its advantage at 4-bit precision.

## F    WHY CSQ IS ALWAYS BETTER THAN CLQ AT 2-BIT

In the weight quantization error experiment of Section 6.4, optimizing for minimum quantization error faces the challenge of maximizing the utility of limited quantization levels. The utility is max-

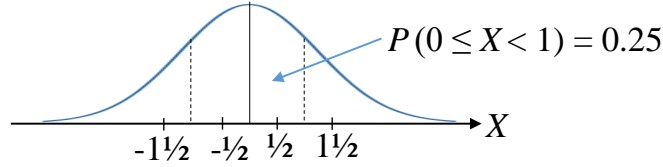

Figure 7: With 2-bit CSQ, quantization levels (shown on the $x$-axis) can resemble Gaussian distribution while, at the same time, real-valued weight data are also uniformly mapped to them (scale parameter $s$ is omitted for brevity).

Table 9: Comparison of CSQ vs. CLQ using knowledge distillation training on ImageNet.

| | Top-1 Accuracy @ Precision | | |
|---|---|---|---|
| **Network** | ResNet-18 | | |
| | *Full Precision: 70.52* | | |
| **Precision (W/A)** | 2/2 | 3/3 | 4/4 |
| CLQ (LSQ) + Knowledge Distillation | 66.99 | 69.77 | **70.63** |
| CSQ + Knowledge Distillation (ours) | **67.24** | **69.90** | 70.56 |

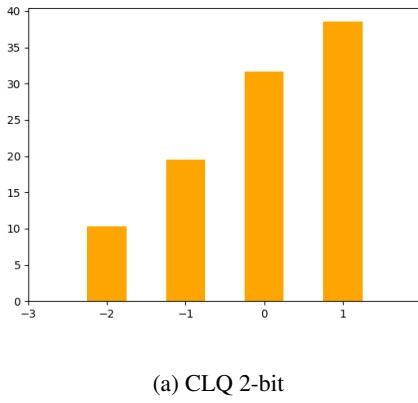

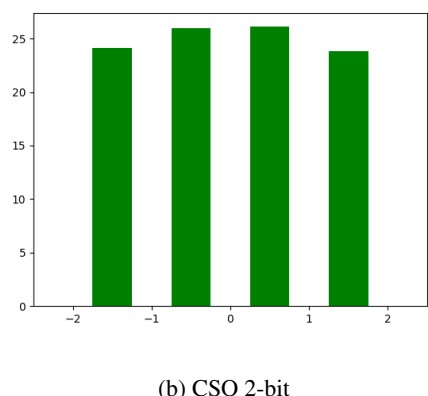

(a) CLQ 2-bit

(b) CSQ 2-bit

Figure 8: How real-valued weight data are mapped to CLQ vs. CSQ after training ResNet-18 on ImageNet.

imized (i) if each quantization level has equal number of real values mapped to it, in the same way as the Lloyd-Max quantization (Max, 1960) is optimal. Also it can be helped a lot (ii) if the quantization levels have the same distribution as the underlying data. In other words, for the first objective the underlying data (when mapped to quantization levels) should be distributed as uniformly as possible, while for the second objective the distribution of quantization levels should resemble that of the underlying data (e.g., Gaussian). This usually creates conflicting requirements, but not in the case of 2-bit CSQ. At 2-bit, CSQ has only three quantization thresholds, $\{-1, 0, 1\}$, and therefore can satisfy both requirements simultaneously: (i) real-valued weight data are uniformly distributed across quantization levels, and at the same time, (ii) quantization levels follow Gaussian (or any symmetric) distribution, as illustrated in Figure 7. Note that this is not possible for 2-bit CLQ, as in other precision for either CLQ or CSQ, which explains why CSQ always shows better performance than CLQ at 2-bit. From the figure, the scale parameter can be determined as $s = \Phi^{-1}(0.75)$, where $\Phi$ is the CDF of standard normal distribution, since $P(0 \leq X < s) = 0.25$ when $X \sim N(0, 1)$). Also our experimental results confirm that indeed real-valued weight data are mapped uniformly to 2-bit CSQ quantization levels as shown in Figure 8.

## G  EXHAUSTIVE SEARCH METHOD FOR QUANTIZATION ERROR EXPERIMENTS

We present quantization error experiments in Section 6.4. To find the step size that minimizes the Quantization Error we use an exhaustive search method. Our exhaustive search method is very similar to the method used by Esser et al. (2019) for their quantization error experiments. The exhaustive search goes as follows. First, we initialize step sizes as:

$$s_0 = \frac{\langle |w| \rangle}{2^p - 1} \tag{23}$$

where $w$ represents full precision weights, $\langle . \rangle$ represents the mean operation, and $p$ is the bit-width. Then for the search space $S = \{0.01s_0, 0.02s_0, 0.03s_0, \cdots, 5s_0\}$, we exhaustively find the value of $s \in S$ that minimizes the target quantization error metric. This helps us find the minimum quan-

tization error using CLQ and CSQ, for any given bit-width. Experimental results and analysis has been presented in Section 6.4. https://www.overleaf.com/project/608a0e19c95387366f0de2f6

# H    MORE ABOUT ZERO-POINT

The difference between $z_{\mathrm{CLQ}}$ and $z_{\mathrm{CSQ}}$ is constant at 0.5 (see (11)) while the possible range of values that can be taken by the quantizer grows exponentially with $b$. Thus we can give the percentage difference between zero-points of CLQ vs. CSQ relative to the entire quantization range by:

$$D = \frac{z_{\mathrm{CLQ}} - z_{\mathrm{CSQ}}}{2^b} \times 100 = \frac{1}{2^{b+1}} \times 100 \tag{24}$$

This equation implies that $D$ decreases as we increase the precision. For example, at 2-bit precision, $D$ is 12.5% but at 4-bit precision, $D$ reduces to mere 3.125%. This shows that at higher precision the difference between CSQ and CLQ becomes negligible compared to the entire distribution range and they should provide similar performance. This is consistent with our experimental results in Section 6.

## H.1    ZERO REPRESENTATION

CSQ does not provide an exact representation for zero. Instead, zero or values slightly less than zero are rounded to $-0.5$ while values slightly greater than zero are rounded to $0.5$ (before scale factor). Despite this it has been shown in Section 6.4 that CSQ can reduce quantization error compared with CLQ. Furthermore, our experimental results have also shown superior performance with CSQ. This serves as an evidence that exact zero representation is not critical for weight quantization, especially at low precision ($< 4$ bits).

## H.2    ZERO PADDING

Zero padding is needed for activation only, not for weight. In case, if zero padding were used for weight as well, we can simulate the exact effect of rounding zeros during QAT, and adjust weight accordingly. So it would not contribute to any performance degradation.

Now, vector and tensor processors (*e.g.* TPU) must process an array of values together, and may "fill" some elements with zeros as needed. This *zero filling* is needed for both weight and activation. For CSQ weight, inexact zero representation may introduce an error or discrepancy between algorithm and realization. This error can be eliminated or minimized, depending on hardware dataflow, by resetting corresponding activation values to zeroes or filling with both $+0.5$ and $-0.5$.

## H.3    WHY PROPOSE CSQ FOR WEIGHT ONLY

CSQ can be used for signed activation. However, activation frequently involves zero padding. Therefore, activation quantization strictly demands exact zero representation to support zero padding. Furthermore Section 4.1 shows that when using CSQ for weight, using $\mathrm{CLQ}_s$ for activation is much better in terms of representational capacity than using CSQ for both weight and activation. To summarize, using CLQ for activation is not only important to support zero padding, but also for best performance. Thus we recommend that using CSQ for weight quantization and CLQ for activation quantization, which provides the best performance with no hardware overhead.

# I    EFFICIENT HARDWARE REALIZATION OF CSQ

## I.1    DIGITAL HARDWARE IMPLEMENTATION

CSQ can be efficiently implemented in hardware and software. To avoid using fractional numbers, in this section we assume the $2\times$ scaled version of CSQ (see Table 1). We can think of CSQ as an $n$-bit extension of BNN encoding. We will show in the section below how we use the XNOR-popcount based inner-product method of BNNs to realize the product of two CSQ vectors. Conventional BNN hardware takes two 1-bit CSQ bit-vectors as an input. Assuming two $N$-element bit-vectors $\mathbf{v}$ and

$\mathbf{x}$, the inner-product is calculated as follows:

$$\mathbf{v}_{csq} \cdot \mathbf{x}_{csq} = 2 \cdot \text{popcount}(\text{XNOR}(\mathbf{v}_{csq}, \mathbf{x}_{csq})) - N \tag{25}$$

In the case of the product between CSQ and unsigned CLQ, we should ignore the values of CSQ when $CLQ_u$ is 0. Therefore, we subtract the number of CSQ values after the popcount operation, which is identical to $\text{popcount}(\mathbf{x}_{clq_u})$.

$$\mathbf{v}_{csq} \cdot \mathbf{x}_{clq_u} = 2 \cdot \text{popcount}(\text{AND}(\mathbf{v}_{csq}, \mathbf{x}_{clq_u})) - \text{popcount}(\mathbf{x}_{clq_u}) \tag{26}$$

We now extend (26) to the multi-bit case so that the inner-product between CSQ and unsigned CLQ vectors can be performed efficiently. Consider a 2-bit CSQ vector and a 2-bit unsigned CLQ vector of $N$ elements.

$$\mathbf{v}_{csq} = \begin{bmatrix} a_1^{N-1}a_0^{N-1} & a_1^{N-2}a_0^{N-2} & ... & a_1^0a_0^0 \end{bmatrix}, \quad \mathbf{x}_{clq_u} = \begin{bmatrix} b_1^{N-1}b_0^{N-1} & b_1^{N-2}b_0^{N-2} & ... & b_1^0b_0^0 \end{bmatrix} \tag{27}$$

Each vector can be split into two bit-vectors by dividing the lower bits and the higher bits of each element.

$$\mathbf{v}_{H_{csq}} = \begin{bmatrix} a_1^{N-1} & a_1^{N-2} & ... & a_1^0 \end{bmatrix}, \quad \mathbf{v}_{L_{csq}} = \begin{bmatrix} a_0^{N-1} & a_0^{N-2} & ... & a_0^0 \end{bmatrix} \tag{28}$$

$$\mathbf{x}_{H_{clqu}} = \begin{bmatrix} b_1^{N-1} & b_1^{N-2} & ... & b_1^0 \end{bmatrix}, \quad \mathbf{x}_{L_{clqu}} = \begin{bmatrix} b_0^{N-1} & b_0^{N-2} & ... & b_0^0 \end{bmatrix} \tag{29}$$

The product of $\mathbf{v}$ and $\mathbf{x}$ can be computed as follows (<< is the bitwise shift-left operation):

$$\mathbf{v} \cdot \mathbf{x} = (\mathbf{v}_H \cdot \mathbf{x}_H << 2) + (\mathbf{v}_H \cdot \mathbf{x}_L << 1) + (\mathbf{v}_L \cdot \mathbf{x}_H << 1) + \mathbf{v}_L \cdot \mathbf{x}_L \tag{30}$$

Each product on the right-hand side of (30) can be computed on the proposed hardware in a single cycle. Thus $\mathbf{v} \cdot \mathbf{x}$ can be computed in four cycles, using an additional adder/accumulator.

### I.2 FAST GPU IMPLEMENTATION

It is possible to speed up BNNs on GPU by using bitwise operations (Hubara et al., 2016), through a custom GPU kernel to boost matrix multiplication on GPU. We have implemented a custom kernel to compute the inner product of two ($N$-dimensional) vectors of $n$-bit CSQ numbers.

$$\mathbf{a} = \begin{bmatrix} a_{n-1}^{N-1}a_{n-2}^{N-1}\cdots a_0^{N-1} & a_{n-1}^{N-2}a_{n-2}^{N-2}\cdots a_0^{N-2} & \cdots & a_{n-1}^0a_{n-2}^0\cdots a_0^0 \end{bmatrix} \tag{31}$$

$$\mathbf{b} = \begin{bmatrix} b_{n-1}^{N-1}b_{n-2}^{N-1}\cdots b_0^{N-1} & b_{n-1}^{N-2}b_{n-2}^{N-2}\cdots b_0^{N-2} & \cdots & b_{n-1}^0b_{n-2}^0\cdots b_0^0 \end{bmatrix} \tag{32}$$

The kernel first concatenates the most significant bits into a single $N$-bit number (if $N > 32$, we break it to a multiple of 32). Repeating this for all $n$ bits, each vector is converted to an $n$-element array.

$$\mathbf{a}_c = \begin{bmatrix} a_{n-1}^{N-1}a_{n-1}^{N-2}...a_{n-1}^0 & a_{n-2}^{N-1}a_{n-2}^{N-2}\cdots a_{n-2}^0 & \cdots & a_0^{N-1}a_0^{N-2}\cdots a_0^0 \end{bmatrix} \tag{33}$$

$$\mathbf{b}_c = \begin{bmatrix} b_{n-1}^{N-1}b_{n-1}^{N-2}\cdots b_{n-1}^0 & b_{n-2}^{N-1}b_{n-2}^{N-2}\cdots b_{n-2}^0 & \cdots & b_0^{N-1}b_0^{N-2}\cdots b_0^0 \end{bmatrix} \tag{34}$$

Then we perform the XNOR-popcount operation $n^2$ times, accumulating its result.

$$s \leftarrow s + \{\text{popcount}(\text{XNOR}(\mathbf{a}_c[i], \mathbf{b}_c[j])) << (i+j)\} \tag{35}$$

