# OpenReview forum: "CSQ: Centered Symmetric Quantization for Extremely Low Bit Neural Networks"
_ICLR.cc/2022/Conference — ICLR 2022 Submitted_

### Official Review · Reviewer_PPJy · 2021-11-01

**Correctness:** 2
**Technical Novelty And Significance:** 2
**Empirical Novelty And Significance:** 2
**Recommendation:** 3
**Confidence:** 4

**Main Review:**

This work's main idea is well-written and pretty straightforward by comparing the proposed CSQ with CLQ (Conventional Linear Quantizer) and explaining that some quantization representations are wasted in the conventional quantization methods. The idea of using a zero-centered quantizer is ok, but it might need more novelty because it has already been used in several previous works. That's why the authors claim that this paper has other contributions in that they propose an explicit quantizer function for CSQ and provide some analysis on the effectiveness of such a scheme at very low precision. However, it is not clear to the reviewer if their claim is technically sound because theoretical analysis or empirical evidence does not support their main contributions strongly. Moreover, there are some cons to demonstrating its effectiveness in the experimental results. Here are a few details regarding this concern.

1. Apart from explaining the problem of CLQ and the expressive power of existing quantization methods, the reviewer is unsure about the necessity of allocating symmetric quantization levels. The learned weight distribution as described in Figure 2 of DFQ [1] may not be symmetrical. Because the weight distribution being learned may be similar to the normal distribution but not a perfectly symmetric structure, allocating symmetric quantization levels need not be assigned to the positive and negative regions. Previous QAT studies such as clipping methods [2], QIL [3], or LSQ [4] have been attempted to reduce the dynamic range of non-uniform quantization or to learn the quantization intervals for this reason. It seems that applying CSQ is not always better than the previous approaches in the above situation. The author should explain first the necessity of allocating symmetric quantization levels by comparing these methods to claim the motivation of this study. Is there any particular reason why the authors excluded non-uniform quantization methods or other QAT approaches to solve the same problem from the discussion?

2. In the experiment, it is explained that CSQ is applied only to weight quantization. However, it doesn't seem easy to conclude that the results presented in Table 5, Table 6, and Figure 3 are entirely caused by symmetrical quantization of CSQ. Nevertheless, it should be possible to clearly explain that CSQ majorly contributes to these results, excluding the effects of activation quantization and learning weights.

3. Overall, the improvement due to CSQ seems to be minor and insignificant for both CIFAR-10 and ImageNet because there is little performance gap when looking at the results presented in Table 5 and Table 6 and results of applying PTQ covered in the appendix. Therefore, it seems not convincing to validate the effectiveness of the proposed CSQ scheme.

4. Some recent works (Boo et al.(2021), Lee et al.(2021), Chen et al.(2021)) have used a quantizer that results in zero-centered quantization levels similar to the one proposed in this paper. So why not compare the experimental results with them?

Minor Comments:
1. The second equality in equation (12) is not intuitive, so it needs to be proved.
2. Equation (30) on page 6 should be replaced with equation (14).
3. There are a few typos.

References:
[1] M. Nagel, M. V. Baalen, T. Blankevoort, and M. Welling, "Data-free quantization through weight equalization and bias correction," ICCV 2019.
[2] R. Banner, Y. Nahshan, E. Hoffer, and D. Soudry, "ACIQ: Analytical Clipping for Integer Quantization of Neural Networks," arXiv e-print, arXiv:1810.05723, Oct 2018.
[3] S. Jung, C. Son, S. Lee, J. Son, J. Han, Y. Kwak, S. Hwang, and C. Choi, "Learning to quantize deep networks by optimizing quantization intervals with task loss," CVPR 2019.
[4] S. Esser, J. McKinstry, D. Bablani, R. Appuswamy, and D. Modha, "Learned step size quantization," ICLR 2019.


**Summary Of The Paper:**

This paper points out that conventional uniform quantization methods do not allocate the same quantization level in the positive and negative areas and proposes Centered Symmetric Quantizer (CSQ), a zero-centered symmetric quantizer, to address the resulting degradation at very low precision. It also presents a method of mapping the proposed CSQ to BNN hardware through bit-wise operations. This paper claims that no specific analysis of CSQ has been made in previous studies and that low-precision quantization results in poor performance due to the allocation of asymmetric quantization levels in the positive and negative sides of the distribution.

**Summary Of The Review:**

To sum up, this paper seems to have several technical flaws for the claims to be well supported.

---

> ### Author Response · Authors · 2021-11-20
> **Reponse to Reviewer PPjy (Part 1/2)**
>
> Thank you for your review. We would like to clarify that an in-depth study and analysis of comparison between CSQ and CLQ is not our only contribution. We also propose a method for hardware realization of CSQ. This contribution is incredibly significant and one of the defining features of our work. The goal of quantization is eventually hardware acceleration. However, none of the works that use a similar quantization method (mentioned in Section 2.3) have presented any consideration for hardware realization. We have shown in our work that it is not a trivial task and proposed a novel method for hardware realization as well. We have tried our best to clarify your concerns and misconceptions about our experiments and analysis.
>
>
> Q1. ”the reviewer is unsure about the necessity of allocating symmetric quantization levels. The learned weight distribution as described in Figure 2 of DFQ may not be symmetrical”
>
> A1. First, one way to deal with asymmetrical weight distribution is to use affine quantization, which however has higher hardware complexity than CLQ or CSQ.
>
> In addition, the fact that weights may not be perfectly symmetric raises the questions like ”Whether a perfectly symmetric quantization (CSQ) actually provides the best representation or a slightly asymmetric quantizer provides better representation”, ”At what precision does CSQ provide better representation”, ”If weights are not perfectly symmetric, why would CSQ perform better than CLQ”, etc. This is precisely why our work has a high significance, and the motivation of our work is to provide answers to questions like these. We have presented the analysis using affine quantization in Section 6.5, which shows that even though ideal quantization levels are not perfectly symmetric, CSQ provides a much better approximation compared to conventional linear quantization (CLQ) especially at lower precision. Furthermore, our experimental results using BRECQ in Table 8 compare CSQ with affine quantization which can learn the asymmetry in weight distribution with some hardware overhead. CSQ even outperforms affine quantizer at 2-bit precision. We also show in Fig. 3 that CSQ can represent the latent distribution with significantly lower quantization error. We have provided sufficient analysis and empirical evidence in Figs. 3 and 4 and Table 3 to support that CSQ provides better representation than CLQ at lower precision, and to answer such questions as why CSQ provides better representation and at what precision CSQ loses its advantage over CLQ.
>
>
> Q.2 ”Previous QAT studies such as clipping methods [2], QIL [3], or LSQ have been attempted to reduce the dynamic range of non-uniform quantization or to learn the quantization intervals for this reason. It seems that applying CSQ is not always better than the previous approaches in the above situation”
>
>
> A2. First, unlike the reviewer claims, those previous QAT papers cited are for **uniform quantization**, and not for **non-uniform quantization**.
>
> Second, those previous approaches are orthogonal to CSQ. In other words, CSQ can be applied in combination with them. In fact, we do use CSQ in conjunction with LSQ, which is compared against using CLQ in conjunction with LSQ. Again, CSQ and CLQ are about quantization levels or quantizer functions, whereas LSQ is about a training method.
>
> Third, we do perform comparison with those previous work (please refer to Table 6 and explanation in Section 6.3). We have already compared against LSQ [1] (CLQ=LSQ) and shown that our results are superior. Our results our substantially better than QAT papers referred by the reviewer such as QIL [4] and ACIQ [3]. To clarify this, we have added some more related works for comparison in Table 6 including QIL.
>
>
> Q3. “Is there any particular reason why the authors excluded non-uniform quantization methods or other QAT approaches to solve the same problem from the discussion?”
>
>
> As clarified, we have already compared results with SoTA QAT methods such as LSQ. While non-uniform quantization can be more powerful than uniform quantization, it incurs some hardware overhead (or at least radically different hardware) as well. Therefore, most of the uniform quantization works such as LSQ and EWGS [2] do not compare their works with non-uniform quantization.

---

> ### Author Response · Authors · 2021-11-20
> **Reponse to Reviewer PPjy (Part 2/2)**
>
> Q4. ”In the experiment, it is explained that CSQ is applied only to weight quantization.”
>
>
> A4. We would like to clarify that activations are quantized in our experiments as well which is shown in Table 6 and explained in Section 6.3 (e.g 2/2 represents 2-bit weight/2-bit activation). Since we are applying CSQ to weights only, activations are quantized using CLQ (signed or unsigned both).
>
> We have justified why CSQ is not applied to activation quantization in Appendix G3 (of original submission). CSQ is a type of uniform ”symmetric” quantization which is used to quantize signed distributions. Activations are usually unsigned due to ReLU and do not require a symmetric quantizer. Furthermore, activations often involve zero-padding which requires exact zero-representation and representational capacity statistics in Table 3 show that CLQ for activations provides higher representational capacity as well. That is why we have proposed and studied CSQ for weights distribution only and we use CLQ for activations throughout our experiments in Tables 5 and 6.
>
>
> Q5. ”it doesn’t seem easy to conclude that the results presented in Table 5, 6, and
> Figure 3 are entirely caused by symmetrical quantization of CSQ”
>
>
> A5. We respectfully disagree. In all of the mentioned experiments (i.e., Table 5, 6) both CSQ and CLQ have 2-bit precision for both weights and activations. The only difference between CLQ and CSQ in the experiments is the choice of weight quantizer while all the other setups are same including activation quantizer and training parameters. There is no factor other than CSQ vs CLQ that could cause the performance improvement in Table 5, 6.
>
> In Fig. 3 we provide an analysis on each quantizer (CLQ and CSQ), to show the quantization error resulting exclusively from the choice of quantizer function. This should conclusively prove that CSQ is the only element contributing to the performance gain. We do not think there is any room to deduce ambiguity or a secondary source of performance improvement.
>
>
> Q6. ”Overall, the improvement due to CSQ seems to be minor and insignificant”
>
>
> A6. Accuracy improvement may seem small. However, since the performance gap with full precision has greatly reduced over the recent years, this improvement is indeed significant. For example, EWGS [2] shows 0.3% improvement at 2-bit precision. It should be noted that accuracy improvement is achieved simply by choosing a quantizer function which provides a different set of quantization levels *for weight quantization only*. Considering this, we achieve a performance improvement of 0.3% consistently across all networks, datasets, and training methods at 2-bit precision, which should be considered significant.
>
>
>
> Q7. ”Some recent works (Boo et al.(2021), Lee et al.(2021), Chen et al.(2021)) have used a quantizer that results in zero-centered quantization levels similar to the one proposed in this paper. So why not compare the experimental results with them?”
>
> First, the objective of this work is not to propose zero-centered quantizer (CSQ) for the first time, but rather to establish and demonstrate that the family of zero-centered quantizers performs superior compared to the conventional quantizers (CLQ), which has never been done before. Therefore, we compare CSQ with CLQ in all experiments.
>
> Second, the papers that have already used zero-centered quantization do not attribute any performance advantage to the choice of zero-centered quantization in their work, but they have used a zero-centered quantization on top of their proposed method. Therefore, it is extremely hard to make a fair comparison with their works because their results may be higher or lower simply based on the superiority of their proposed (other) methods. We have also mentioned this in our paper in Section 6.3.
>
>
>
> [1] Esser, Steven K., et al. ”Learned step size quantization.” arXiv preprint arXiv:1902.08153 (2019).
>
>
> [2] Lee, Junghyup, Dohyung Kim, and Bumsub Ham. ”Network Quantization with Element-wise Gradient Scaling.” Proceedings of the IEEE/CVF Conference on Computer Vision and Pattern Recognition. 2021.
>
>
> [3] Banner, Ron, et al. ”Aciq: Analytical clipping for integer quantization of neural networks.” (2018).
>
>
> [4] Jung, Sangil, et al. ”Learning to quantize deep networks by optimizing quantization intervals with task loss.” Proceedings of the IEEE/CVF Conference on Computer Vision and Pattern Recognition. 2019.

---

### Official Review · Reviewer_Z3wR · 2021-11-02

**Correctness:** 3
**Technical Novelty And Significance:** 2
**Empirical Novelty And Significance:** Not applicable
**Recommendation:** 5
**Confidence:** 5

**Main Review:**

This paper seems to be the first to deep dive into the difference between CLQ and CSQ, although these quantization techniques are not new. However, the resulting analysis seems to be straightforward for the following reasons:

- The conclusion that CSQ could be (marginally) better seems to be as expected, since CLQ might not be efficient for representing the signed (or centered) distribution. Although this paper explicitly revealed this fact, it does not seem to provide new insights.

- The analysis based on the representation capacity seems to be weak. Table 3 reveals that the number of distinct output states varies for different combinations of quantizers. Therefore, it does not necessarily prioritize CSQ to CLQ. Furthermore, the importance of the representation capacity is not clear; although CSQ-CLQs has a 50% larger representation capacity for the signed activation, its accuracy gain seems to be marginal (~0.4% from Table 5,6).

- The accuracy results reported in Table 6 seem to be misleading. 1) The authors did not put the accuracy results of LSQ (= CLQ case) for ResNet18 and ResNet34 (67.6% and 71.6%, respectively), which are significantly higher than CSQ. 2) In the case of MobileNetV2, the reported accuracy results CSQ seem to be significantly lower than the state-of-the-art (e.g., PROFIT(ECCV2020) 4-bit MobileNet-V2 = 69.06% compared to CSQ=66.98%).



**Summary Of The Paper:**

This paper analyzed the difference in quantization methods between the conventional linear quantizer (CLQ) and centered symmetric quantization (CSQ). The authors explained that CSQ might exploit the limited quantization states better to represent the data than CLQ, with supporting analysis on representation capacity. The authors further proposed a bitwise implementation of CLQ and CSQ. The evaluation in terms of the quantization accuracy and GPU implementation speed is given for their analysis.

**Summary Of The Review:**

This paper seems to provide a better understanding of the low-precision (= 2-bit) quantizers, but the discovery itself seems to be marginally significant. Therefore, I am inclined toward rejection.

---

> ### Author Response · Authors · 2021-11-20
> **Response to Reviewer Z3wR (Part 1/2)**
>
> Thank you for your review. We would like to clarify that an in-depth study of comparison between CSQ and CLQ is not our only contribution. We also propose a method for hardware realization of CSQ. This contribution is incredibly significant and one of the defining features of our work. The goal of quantization is eventually hardware acceleration. However, none of the works that use a similar quantization method (mentioned in Section 2.3) have presented any consideration for hardware realization. We have shown in our work that it is not a trivial task, and proposed a novel method for hardware realization as well.
>
> We have tried to address your questions and concerns regarding the experiments and analyses.
>
>
> Q1. “The conclusion that CSQ could be (marginally) better seems to be as expected”
>
>
> A1. It is not a given that CSQ would be better. In fact, the contrary can be argued as well. The weight distribution does not always follow an exact zero-mean Gaussian distribution but there is usually a slight asymmetry; therefore, it can be argued that the asymmetry in CLQ can be advantageous to represent slightly asymmetric distribution.
>
> Also, there are important questions regarding CSQ vs CLQ that no previous work answers, such as “Can a perfectly zero-centered quantization (CSQ) provide better representation than slightly asymmetric quantization (CLQ)?”, ”At what precision does a perfectly symmetric quantization (CSQ) start to give better representation & accuracy?”, etc. In this work we have thoroughly explored and answered these questions with significant empirical evidence. We would like to refer to Fig. 4 of the paper, which clearly shows that at higher precision the ideal zero-point does not converge to a perfect symmetry. Furthermore, the experimental results on BRECQ [4] in Table 8, which uses an affine quantizer, shows that at 3- and 4-bit precision a learn-able asymmetry (affine quantization) performs superior to CSQ. All of this evidence conclusively shows that it cannot be expected as a common knowledge that CSQ would be better. This work provides important insights regarding the choice of quantization levels for different precisions, along with their limitations and advantages.
>
>
> Q2. ”Table 3 reveals that the number of distinct output states varies for different combinations of quantizers. Therefore, it does not necessarily prioritize CSQ to CLQ”
>
>
> A2. The reviewer has overlooked an important fact that we are proposing CSQ for **weight** quantization, not for activation. In other words, we are **not** claiming that CSQ is always better than CLQ. The important contribution of this paper is to analyze the quantitative advantage of CSQ over CLQ and the limitations of CSQ, which no previous work has provided. Indeed, results such as Table 3 are exactly why this paper may be interesting to the readers or attendees of this conference.
>
>
> Q3. ”importance of the representation capacity is not clear; although CSQ-CLQs has a 50% larger representation capacity for the signed activation, its accuracy gain seems to be marginal”
>
>
> A3. First, in our experimental settings we use ReLU activation function, which gives unsigned activations. Therefore, we should look at the left side of the table, where CSQ-CLQ_u (unsigned activation) has 22.2% higher representational capacity than CLQ_s-CLQ_u at 2-bit. The right side showing that CSQ-CLQ_s (signed activation) has 50% higher capacity at 2-bit, is given for completeness.
>
> Bi-Real Net [3] increases the representational capacity exponentially and their results show only 5% improvement. In comparison, our 22.2% increase in representational capacity is relatively small and the performance gain is proportional. It is important to clarify that the objective of this work is not so much on improving the representational capacity of a quantized network, but more on investigating **why** and **when** CSQ performs superior to CLQ.

---

> ### Author Response · Authors · 2021-11-20
> **Response to Reviewer Z3wR (Part 2/2)**
>
> Q4. ”The authors did not put the accuracy results of LSQ (= CLQ case) for ResNet18 and ResNet34 (67.6% and 71.6%, respectively)”
>
>
> A4. The accuracy results shared by LSQ [1] are for pre-activation ResNet which is why they are higher. Since the code is not publicly available, we implemented LSQ ourselves on standard ResNet. This is mentioned in Section 6.3 as well. This is consistent with the methodology of other previous works, e.g., LSQ+ [5] and EWGS [2], which have also compared with LSQ on standard ResNet with performance lower than the reported performance in the LSQ paper.
>
>
>
> Q5. ”In the case of MobileNetV2, the reported accuracy results CSQ seem to be significantly lower than the state-of-the-art (e.g., PROFIT(ECCV2020) 4-bit MobileNet-V2 = 69.06% compared to CSQ=66.98%)”
>
> A5. This is because PROFIT uses an advanced training method specifically proposed to train quantized MobileNet. We have demonstrated our results using LSQ [1] which is a SoTA training method generalizable to all networks. Since the objective of our work was to study the choice of quantization levels using CSQ and CLQ, and not the training method for quantization-aware training, we used LSQ to report results on MobileNet as well.
>
> [1] Esser, Steven K., et al. ”Learned step size quantization.” arXiv preprint arXiv:1902.08153 (2019).
>
> [2] Lee, Junghyup, Dohyung Kim, and Bumsub Ham. ”Network Quantization with Element-wise Gradient Scaling.” Proceedings of the IEEE/CVF Conference on Computer Vision and Pattern Recognition. 2021.
>
> [3] Liu, Zechun, et al. ”Bi-real net: Enhancing the performance of 1-bit cnns with improved representational capability and advanced training algorithm.” Proceedings of the European conference on computer vision (ECCV). 2018.
>
> [4] Li, Yuhang, et al. ”Brecq: Pushing the limit of post-training quantization by block reconstruction.” arXiv preprint arXiv:2102.05426 (2021).
>
> [5] Bhalgat, Yash, et al. ”Lsq+: Improving low-bit quantization through learnable offsets and better initialization.” Proceedings of the IEEE/CVF Conference on Computer Vision and Pattern Recognition Workshops. 2020.

---

### Official Review · Reviewer_hDhb · 2021-11-03

**Correctness:** 3
**Technical Novelty And Significance:** 2
**Empirical Novelty And Significance:** 2
**Recommendation:** 5
**Confidence:** 4

**Main Review:**

Strength: A new idea of applying centered symmetric quantizer for model quantization.

Weakness:

1. Speedup setting: The authors only compare the single matrix multiplication runtime, which does not necessarily represent the speedup and the accuracy of the whole network [1].

2. Due to the new quantization level, the authors are encouraged to conduct experiments on other tasks (e.g., detection and NLP tasks.), which can present the generalization and robustness of CSQ.

3. The accuracy improvement is marginal than CLQ on the ImageNet dataset.

4. If people optimize sparse mand quantized models jointly [2, 3], is the CLQ with sparsity equal to CSQ?


[1]. MQBench: Towards Reproducible and Deployable Model Quantization Benchmark

[2]. Compressing Deep Neural Networks with Pruning, Trained

[3]. A unified framework of dnn weight pruning and weight clustering/quantization using admm

**Summary Of The Paper:**

This paper presents a centred symmetric quantizer(CSQ), a new symmetrical quantization scheme.

**Summary Of The Review:**

The analysis and experiments have some issues (see weakness). I will give a borderline due to those concerns and change it accordingly based on rebuttal.

---

> ### Author Response · Authors · 2021-11-20
> **Response to Reviewer hDhb**
>
> Thank you for your review. We have tried to clarify confusion and address your concerns.
>
> Q1. ”The authors only compare the single matrix multiplication runtime, which does not necessarily represent the speedup and the accuracy of the whole network [1].”
>
> A1. It is true that we have compared the single matrix multiplication runtime, but it is because everything is exactly the same between CLQ and CSQ outside of the matrix multiplication kernel, including the number of bits needed. Our claim is that CSQ has no runtime overhead compared with CLQ of the same precision, which is supported by our matrix multiplication comparison experiments. If we claimed some **speedup** by our method (which is **not** the case in our paper), we would need to compare the whole network.
>
> Regarding accuracy, we do evaluate and compare the whole networks. Any performance degradation in quantized networks due to other operations pointed out by MQBench (such as batch normalization folding) is independent of the choice of a quantizer function and would be consistent with previous works as well. Therefore, full network realization is unnecessary for the scope of our paper.
>
> Now, MQBench [4] (cited by the reviewer) focuses on full network hardware realization and points out some practical problems in hardware deployment. However, the focus of our paper is on the choice of a quantizer function and quantization levels, and its efficient hardware realization. (Although some previous works have used quantizer functions similar to ours (e.g., [2], [5], [6], etc.; refer to Section 2.3 for detail), they have not considered the hardware realization of the quantized representation.)
>
>
>
> Q2.”the authors are encouraged to conduct experiments on other tasks (e.g., detection and NLP tasks.), which can present the generalization and robustness of CSQ”
>
>
> A2. We thank the reviewer for the suggestion. We agree that evaluating CSQ on other tasks can help. However, in our paper we have focused on revealing the differences between CSQ and CLQ, showing its limitations and its effectiveness at different precisions as well as hardware realization. To demonstrate generalization ability of our method, we have shown experimental results on both **quantization-aware training** (in Tables 5 and 6, using SoTA method LSQ [1]) and **post-training quantization** (in Table 8, using SoTA method BRECQ [3]).
>
>
> Q3. ”The accuracy improvement is marginal than CLQ on the ImageNet dataset.”
>
>
> A3. Accuracy improvement may seem small. However, since the performance gap with a full-precision network has greatly reduced over the recent years, this improvement is indeed significant. For instance, EWGS [2] shows only 0.3% improvement at 2-bit precision on ResNet-18 ImageNet than the previous state-of-the-art work. In addition, it should be noted that our accuracy improvement is obtained for free, simply by modifying the quantization levels of quantized weights.
>
>
> Q4. ”If people optimize sparse and quantized models jointly [2, 3], is the CLQ with sparsity equal to CSQ”
>
>
> A4. No, CSQ and CLQ will not be equal even if joint sparsity-and-quantization optimization is performed. Sparsity optimization is understood as increasing the number of zero values, which can lead to smaller model size and possibly faster inference speed as well.
>
> CLQ: CLQ already has zero values in its quantized representation (see Table 1), therefore, joint quantization-and-sparsity optimization would be equivalent to quantizing a greater number of weight parameters to ’zero’ quantization levels. Hence, it will never be equal to CSQ. (Recall that CSQ does not use ‘zero’ quantization level.)
>
> CSQ: CSQ does not have zero value in its quantization levels. In this case sparsity optimization is either impossible or can be interpreted as using one of the existing quantization levels a lot more so that we could have model size reduction benefit. Either way, it is clear that sparsity optimized model will never be the same as using CLQ.
>
>
>
> [1] Esser, Steven K., et al. ”Learned step size quantization.” arXiv preprint arXiv:1902.08153 (2019).
>
>
> [2] Lee, Junghyup, Dohyung Kim, and Bumsub Ham. ”Network Quantization with Element-wise Gradient Scaling.” Proceedings of the IEEE/CVF Conference on Computer Vision and Pattern Recognition. 2021.
>
>
> [3] Li, Yuhang, et al. ”Brecq: Pushing the limit of post-training quantization by block reconstruction.” arXiv preprint arXiv:2102.05426 (2021).
>
>
> [4] Li, Yuhang, et al. ”MQBench: Towards Reproducible and Deployable Model Quantization Benchmark.” (2021).
>
>
> [5] Choi, Jungwook, et al. ”Bridging the accuracy gap for 2-bit quantized neural networks (qnn).” arXiv preprint arXiv:1807.06964 (2018).
>
>
> [6] Chen, Peng, et al. ”Aqd: Towards accurate quantized object detection.” Proceedings of the IEEE/CVF Conference on Computer Vision and Pattern Recognition. 2021.

---

### Official Review · Reviewer_FpP9 · 2021-11-09

**Correctness:** 3
**Technical Novelty And Significance:** 2
**Empirical Novelty And Significance:** 2
**Recommendation:** 5
**Confidence:** 4

**Details Of Ethics Concerns:**

This paper doesn't have ethical concerns.

**Main Review:**

I think the paper is generally well written.

The authors show that CSQ can be efficiently expressed by binary representations, and then be calculated using XNOR operations. I appreciate the details and analysis provided in the main text and supplementary materials.

CSQ has quantization results for mobilenets, which are compact models that generally are harder for quantization. Experiments on large datasets (such as ImageNet) are also helpful to validate the effectiveness of CSQ.

In addition, some ablation study are meaningful, such as the BRECQ-CSQ experiments in the supplementary material.

Despite the merits, I think the paper still has the following weaknesses:
1. The scope of application is quite limited. From the paper we can see that CSQ can consistently outperform CLQ on 2-bit, but is not guaranteed to be better on 3-bit or 4-bit. Plus, there are other orthogonal methods that can alleviate the quantization degradation on 2-bit. Although the authors claim the difference between CSQ and previous methods (Section 2.3, Stochastic precision ensemble etc.), the novelty of CSQ is not particularly strong.
2. In Table 5, the ESQ results mentioned on the caption are missing.
3. The experimental results of CSQ didn't show great improvement over CLQ. For example, on ImageNet with 2-bit ResNet34, the gain is 0.26%. I wonder if the accuracy gain will remain if advanced quantization-aware training methods (with or without distillation) or mixed-precision methods are applied together with CLQ/CSQ.
4. Some state-of-the-art results need to be compared with, for example:
[1] Distribution-Aware Adaptive Multi-Bit Quantization.
5. The ResNet34 baseline (73.3%) used in the experiment is not state-of-the-art (>74%). I suggest using ResNet50 which is a more common and standard choice.

**Summary Of The Paper:**

This paper presents a centered symmetric quantizer (CSQ) that can map floating point tensors into zero-centered quantized integer values. The CSQ method works well for ultra-low bit quantization such as 2-bit.
The authors also propose a binary coding method to run efficiently on hardware.

**Summary Of The Review:**

I think CSQ is a simple but effective method that can be helpful for ultra-low bit quantization. However, the novelty and application scope are limited and the experiments have room for improvement. Consequently, I think the paper is marginally below the acceptance threshold.

---

> ### Author Response · Authors · 2021-11-20
> **Response to Reviewer FpP9 (Part 1/2)**
>
> We thank you for your review. We are encouraged that you found the details and analysis, our experimental results, and ablation study using BRECQ [6] helpful. We have tried to clarify your concerns and used them to update the paper.
>
> Q1. ”CSQ can consistently outperform CLQ on 2-bit but is not guaranteed to be better on 3-bit or 4-bit.”
>
> A1. Our experiments show that CSQ is consistently better at 2-bit and 3-bit quantization. The confusion that CSQ does not consistently outperform CLQ on 3-bit may be caused by the BRECQ [6] (post-training quantization) experiments in Table 8. However, as we have explained in Appendix D, BRECQ uses affine quantizer instead of CLQ. Affine quantizer is more powerful than CLQ but also requires more hardware resources, whereas CSQ does not.
>
> We have proposed CSQ for low precision quantization only (i.e., 2~3-bit) as mentioned in Section 1 and Abstract. As for 4-bit precision, we have already discussed why CSQ sometimes performs worse than CLQ at 4-bit in Section 6.7. To repeat, at 4-bit precision the representational capacity is not a bottleneck in performance and therefore neither CSQ nor CLQ conclusively outperforms the other.
>
> Q2. ”the novelty of CSQ is not particularly strong”
>
> A2. The contribution and novelty of our work is a detailed study and analysis on the best quantized representation of weight parameters from the perspective of quantization levels. However, this is not the only novelty of this paper. We also proposed a method for hardware realization of CSQ as claimed in Section 1 and explained in Section 5. This contribution is also significant and one of the unique features of our work. Many will agree that one of the primary goals of quantization is hardware acceleration, but none of the works that use a similar quantization method to ours (mentioned in Section 2.3) have presented any consideration for hardware realization. We have shown in our work that efficient hardware realization of CSQ is not trivial, and have proposed a novel method for hardware realization as well.
>
> Q3. ”In Table 5, the ESQ results mentioned on the caption are missing.”
>
> A3. We intended to show the result for ESQ defined in Section 3.2 as an upper bound for performance, but it was later removed partly due to page limit and partly because the result was not always the best (sometimes CSQ was better than ESQ!). ESQ, being sometimes worse than CSQ is not necessarily wrong or incorrect, but it just means that ESQ does not serve any useful purpose in our analysis, so we decided not to include it in the final submission. The mention of ESQ in Table 5 survived by mistake, which is fixed in the revised version.
>
> Q4. ”I wonder if the accuracy gain will remain if advanced quantization-aware training methods (with or without distillation) or mixed-precision methods are applied?”
>
> A4. To address your concerns, we have added experimental results on ResNet-18 using LSQ with knowledge distillation on ImageNet in revision.

---

> > ### Comment · Reviewer_FpP9 · 2021-11-24
> > **Reply to Author's Response**
> >
> > I appreciate the efforts the authors spent on answering my questions. The author's response addresses some of my concerns and I will discuss them with other reviewers and AC to have a fair decision.

---

> ### Author Response · Authors · 2021-11-20
> **Response to Reviewer FpP9 (Part 2/2)**
>
> Q5. ”Some state-of-the-art results need to be compared with, for example: Distribution-Aware Adaptive Multi-Bit Quantization.”
>
> A5. DMBQ [5] uses a non-uniform quantization scheme whereas CSQ is uniform quantization. Non-uniform quantization is expected to perform better as it is not constrained to use uniform step size but uses non-conventional hardware and may result in extra hardware overhead. This has been mentioned in Section 3.2. Therefore, most of the uniform quantization works such as LSQ [1], EWGS [2], etc. do not compare their methods with non-uniform quantization schemes.
>
> Q6. ”The ResNet34 baseline (73.3%) used in the experiment is not state-of-the-art (>74%).”
>
> A6. We have used the official PyTorch trained model as the ResNet-34 baseline. Many previous works such as EWGS [2] and Bi-Real Nets [3] have also reported the same full-precision performance as our work for ResNet-34. Also, DBMQ [5], the paper mentioned by the reviewer in the previous comment, uses 73.7% baseline for ResNet34 (which is <74%). LSQ [1] is the only work we are aware of that uses >74% baseline. However, as pointed out in LSQ+ [4] and EWGS [2] as well as our own paper in Section6.3, LSQ uses **pre-activation ResNet** which has higher accuracy.
>
> [1] Esser, Steven K., et al. ”Learned step size quantization.” arXiv preprint arXiv:1902.08153 (2019).
>
> [2] Lee, Junghyup, Dohyung Kim, and Bumsub Ham. ”Network Quantization with Element-wise Gradient Scaling.” Proceedings of the IEEE/CVF Conference on Computer Vision and Pattern Recognition. 2021.
>
> [3] Liu, Zechun, et al. ”Bi-real net: Enhancing the performance of 1-bit cnns with improved representational capability and advanced training algorithm.” Proceedings of the European conference on computer vision (ECCV). 2018.
>
> [4] Bhalgat, Yash, et al. ”Lsq+: Improving low-bit quantization through learnable offsets and better initialization.” Proceedings of the IEEE/CVF Conference on Computer Vision and Pattern Recognition Workshops. 2020.
>
> [5] Zhao, Sijie, Tao Yue, and Xuemei Hu. ”Distribution-Aware Adaptive Multi-Bit Quantization.” Proceedings of the IEEE/CVF Conference on Computer Vision and Pattern Recognition. 2021. [6] Li, Yuhang, et al. ”Brecq: Pushing the limit of post-training quantization by block reconstruction.” arXiv preprint arXiv:2102.05426 (2021).

---

### Author Response · Authors · 2021-11-20
**General Response**

We thank all reviewers for their valuable feedback. We have tried our best to address the concerns and clarify the confusion through individual responses. Based on the valuable advice and suggestions from the reviewers, we have revised the paper as best as we can. We hope that our revised version will be helpful for the reviewers and other readers to clarify their questions and understand the significance of our work.

We have made the following changes in revision:

1. We have added comparisons to more quantization-aware training methods for ImageNet experiments in Table 6. Additional comparisons are made to clarify the superiority of our work.

2. To address Reviewer FpP9’s question about whether the accuracy improvement will remain if advanced quantization-aware training method with knowledge distillation is applied, we have added some additional experiments using LSQ with knowledge distillation in Appendix E and Table 9.

3. As pointed out by Reviewer FpP9, there is an incorrect reference to ESQ results in Table 5. This has been removed.

4. [Minor] Fixed some mistakes in equation references in Section 5.

---

### Decision · Program_Chairs · 2022-01-20

**Decision:**

Reject

**Comment:**

### Description
The paper investigates the choice of a fixed quantization grid for weights. Namly, the paper observes that symmetric uniform quantization levels such as {-1.5,-0.5,0.5,1.5} lead to better results than non-symmetric ones, e.g. {-2,-1,0,1}. While it is a small thing, it can be appreciated that it is investigated systematically and pedantically, proposing an explanation and showing experimentally that the effect is constantly present in favour of symmetric quantization. While the improvement is small, it comes almost at no cost. A part of the contribution proposes an efficient implementation.

### Decision
Reviewers and AC came to a consensus that the contribution of the paper is marginal. Symmetric quantization schemes themseleves were already employed by many models, albeit without analysis or even a discussion of such choice. The analysis presented in the paper was found unconvincing by the reviewers (see below). The efficient implementation follows from basic linear algebra (see below). The potential impact of the work was considered as limited due to a rather marginal observed improvement. The average rating of the paper was 4.5. Therefore must reject.

### Details
Regarding the proposed analysis of CSQ, it is not clear, why the number of quantization levels of an elementary product matters, given that these numbers are then summed over all corresponding input channels and spatial dimensions of a convolution kernel applied at a single location. It is questionable whether the number of these quantization levels indeed corresponds to the representation capacity. Finally, the paper misses to demonstrate the effect on binary (1 bit) networks. In this case the standard approach is to use {-1,1} weights and {-1,1} activatinos. The paper could investigate the case of {0,1} activations, where there would be 50% more unique possible outputs from the product, namely {-1,0,1} to validate their hypothesis. If the hypothesis holds, an improvement in the binary case would be observed. This is important since the binary case is know to be the hardest and since the respective recommendation of representations would be non-standard. It could be further questioned why the distribution of real-valued weights has any relevance (such as in the arguments in appendix E) if the model is trained from scratch? A training method need not keep any real-valued latent weights in the first place.

The technical part in section 5 "efficient realization" adds very little, if anything, to the paper's contribution. A simple linear algebra suffices to see that

$(W-0.5) \ast x = W* x - 0.5 I \ast x,$

where $I$ is the kernel of ones of the same shape as $W$. It is clear that the convolution $I \ast x$ can be implemented efficiently (e.g. it is just a sum over channels followed by a separable spatial only convolution) and is not a bottleneck and. The final detail such as whether to slice by bits and use popcount for it or to use 8-bit addition, depend very much on the choice of the bit-packed representation and the hardware available. It would be known to engineers in the field how to implement it efficiently.